# A COP28 Perspective: Does Chinese Investment and Fintech Help to Achieve the SDGs of African Economies?

Aimin Zhang [1,*], Moses Nanyun Nankpan [2,*], Bo Zhou [2], Joseph Ato Forson [3], Edmund Nana Kwame Nkrumah [4] and Samuel Evergreen Adjavon [3]

1. School of International Business Communication, Dongbei University of Finance and Economics, 217 Jian Shan Street, Sha He Kou District, Dalian 116025, China
2. School of Public Finance and Taxation, Dongbei University of Finance and Economics, 217 Jian Shan Street, Sha He Kou District, Dalian 116025, China; yourab@163.com
3. Department of Applied Finance and Policy Management, School of Business, University of Education, Winneba P.O. Box 25, Ghana; datoeagle@yahoo.com (J.A.F.); samuelevergreen.adjavon@gmail.com (S.E.A.)
4. Doctoral in Business Administration Unit, Noble International Business School, No. 9 Arko Lane, South Legon, Burma Camp, Accra P.O. Box BC 212, Ghana; nkrumah@nibs.edu.gh
* Correspondence: 19970019@dufe.edu.cn (A.Z.); mnankpan@yahoo.com (M.N.N.)

**Abstract:** Scientific consensus affirms human activity, particularly carbon emissions from market participants, drives global warming. Foreign investment, crucial for sustainability in developing nations, now faces scrutiny regarding its impact on environmental quality in emerging economies. This study examines the influence of Chinese Outward Foreign Direct Investment (OFDI) and fintech on environmental conditions in the top five Chinese-invested African economies, alongside factors such as energy consumption, economic performance, and unemployment affecting $CO_2$ pollution. Quarterly data from 2006–2021 confirm cointegration among variables via panel unit root and cointegration tests. Panel ARDL method estimates coefficients for short and long-run effects. Our findings reveal: (1) A 1% increase in Chinese investment leads to a 0.56% decrease in $CO_2$ emissions, supporting its positive environmental impact. (2) Fintech adoption also demonstrates a beneficial effect, with a 1% increase associated with a 0.18% reduction in $CO_2$ levels. (3) Total energy consumption, as expected, has a detrimental impact, causing a 0.92% increase in $CO_2$ emissions with a 1% rise. (4) Interestingly, economic growth fosters environmental sustainability, while unemployment correlates negatively with it. These findings suggest that targeted Chinese investments and fintech adoption can aid in mitigating $CO_2$ pollution in African economies while balancing economic considerations.

**Keywords:** sustainability; Chinese OFDI; fintech; top 5 Chinese-invested African economies; panel ARDL

## 1. Introduction

Sustainable development goals (17 SDGs) have drawn much attention from regulators, foreign investors, economists, and academic scholars in recent decades. Environmental sustainability is a central issue addressed by the United Nation's Agenda 2030 for Sustainable Development [1]. Many studies have also investigated the causes and consequences of environmental degradation [2]. For example, the Ahmad et al., [3] warned that the economic and social damage caused by global warming will dwarf the losses of both World Wars combined. $CO_2$ pollution, primarily from energy production, is a major driver of this global warming. This has garnered significant attention, with international parties addressing the issue at a roundtable during COP28 [4–6]. To achieve sustainable goals, especially $CO_2$ pollution reduction, studies by several scholars have shown that foreign investment is crucial for facilitating sustainable development [7–9].

In macroeconomic planning, foreign investment plays a positive role in economic growth by generating jobs and fostering innovation in green technologies [10]. This contribution extends beyond direct benefits, indirectly supporting the achievement of sustainable

development goals [11,12]. Less developed economies, particularly the 33 African nations among them, face significant challenges in achieving the Sustainable Development Goals (SDGs) due to their low socioeconomic baseline [13]. Limited financial resources hinder their ability to plan and implement sustainable development initiatives [14]. Additionally, these countries often grapple with war, natural disasters such as droughts, and high levels of corruption, further entrenching their difficulties. However, foreign investment (FDI) presents a potential opportunity for these countries to overcome these obstacles and accelerate progress towards the SDGs.

Though developing countries often view FDI as a potent tool for economic progress, modernisation, income growth, and job creation [15], concerns regarding its potential negative impacts on social and environmental development have arisen [16]. Recognizing these concerns, the concept of Sustainable Investment emerges as a critical force for achieving all 17 Sustainable Development Goals (17-SDGs). While existing research investigates the influence of foreign investment on environmental quality by examining various relationships, such as foreign investment and trade with environmental quality [17], foreign investment and energy consumption with environmental quality [18], and foreign investment and economic growth with environmental pollution [19], a gap remains in the literature specifically of African regions. Based on a comprehensive literature review, no study currently examines the impact of Chinese investment and fintech on $CO_2$ pollution in the top five Chinese-invested African economies (Nigeria, South Africa, Egypt, Angola, Republic of the Congo). This study will address this critical knowledge gap.

The purpose of the study, as well as the contribution it makes, may be broken down into four primary scientific questions, which are organized as follows: (a) To what extent does Chinese foreign investment support the top five Chinese-invested African economies in achieving their climate objectives? (b) How does the utilization of fintech influence the pursuit of sustainable development goals in the top five economies receiving Chinese investment in Africa? (c) Does the total energy consumption pattern in the top five Chinese-invested African economies contribute to improved environmental outcomes? (d) How does the interplay of economic growth and unemployment rates affect environmental degradation in the top five African economies heavily invested in by China? These questions will be addressed using highlighted scientific approaches, and the findings will contribute valuable insights to global discussions on the nexus between energy, economic growth, and environmental conservation, crucial for international stakeholders invested in sustainable development and climate action.

After the introduction, there is a comprehensive literature review in Section 2. Section 3 details the data and methodology, while Section 4 provides and analyses the results with interpretation. Finally, part five concludes with closing remarks and policy framing.

## 2. Literature Review

In economic and finance literature, contrasting views exist regarding the impact of foreign investment on $CO_2$ pollution. Two prominent hypotheses, the pollution haven hypothesis and the pollution halo hypothesis, address this debate. Furthermore, the impact of foreign direct investment (FDI) on the environment differs dramatically between developing and developed economies. This is due to a phenomenon known as carbon offshoring, where developed economies shift highly polluting industries to developing countries. Ref. [20] have extended our understanding by documenting that this allows advanced economies to reduce their $CO_2$ pollution while saddling developing nations with environmental degradation. Two studies have confirmed this impact trend in China and the USA, respectively [21,22]. However, a counterpoint exists in the findings of [23–25] who suggest imports may harm environmental quality in the long run, but FDI does not necessarily worsen it.

The intricate relationship between FDI and environmental well-being remains a subject of debate, with numerous academic works supporting both positive (pollution haven hypothesis) and negative (pollution halo hypothesis) consequences. This complexity

intensifies when examining the context of the China-Africa partnership for two distinct reasons. Firstly, China's pronounced commitment to investing in Africa's renewable energy sector [26,27] raises hopes for a greener, more efficient future. However, concerns emerge due to the common practice of African governments pledging vast natural resources as security for these Chinese investments and trade agreements [28]. This reliance on resource extraction [29] poses a potential threat to both environmental sustainability and long-term economic growth. While past researchers have investigated the overall flow of FDI and trade into Africa [30–32], this study takes a different approach by examining China's investment and fintech from the top five Chinese-invested African economies.

Recent research has begun to explore the environmental consequences of financial technologies, particularly through the lens of energy-intensive digital currencies such as cryptocurrencies. These studies have highlighted the significant electricity (energy) consumption and carbon pollution associated with cryptocurrency mining and use, with some suggesting its environmental impact rivals that of entire countries. For example, ref. [33] famously compared the energy demand of cryptocurrencies to that of Ireland. The meteoric rise of Bitcoin, driven by its decentralized nature and speculative allure, has cast a long shadow on the environment. The insatiable energy demands of its mining apparatus, primarily fueled by carbon-intensive sources like such as coal [34], are unleashing a cascade of ecological woes.

In China, the epicenter of Bitcoin production, coal-fired mining operations pump noxious fumes into the atmosphere, exacerbating pre-existing air pollution and jeopardizing public health [35]. This grim reality extends beyond localized contamination, as Bitcoin's escalating electricity consumption translates into a surge in greenhouse gas emissions. Coal and thermal plants, the unsung heroes of this digital gold rush, spew potent $CO_2$ into the air, fanning the flames of global warming and jeopardizing delicate ecosystems. The consequences are dire, not merely for pristine landscapes but for human health as well. Increased air pollution, a direct outcome of Bitcoin's insatiable hunger for energy, has been linked to a disturbing rise in mortality rates.

Building upon the premise that energy consumption and related carbon footprint are crucial concerns in cryptocurrency [36], a pioneering study investigated the energy disparity among prominent digital assets. Focusing on Bitcoin (BTC), Ethereum (ETH), Litecoin (LTC), and Monero (XMR), the researchers sought to quantify their operational energy demands and associated environmental ramifications. The findings revealed a significant disparity in energy consumption amongst these four currencies, with Bitcoin emerging as the most voracious, exceeding the combined energy expenditures of the other three. Notably, the energy consumption of crypto-mining was also found to surpass traditional extractive industries such as gold and copper mining, highlighting the burgeoning environmental concerns surrounding blockchain technology. Ref. [37] documented that Bitcoin mining emissions alone could push global warming beyond the critical 2 °C threshold. The positive impact of fintech on environmental sustainability is also supported by research, such as [38,39]. Ref. [40] have documented that fintech has a negative impact on environmental pollution.

On the other hand, Fintech acts as a green alchemist, transmuting traditional finance into a digital elixir that shrinks carbon footprints. Its touch breathes life into online transactions, supplanting emission-heavy offline workflows and painting a greener canvas for economic activity. By shifting financial activity online, fintech can become a climate warrior, wielding digital tools to slice $CO_2$ emissions from unnecessary travel associated with traditional finance [41]. Online platforms facilitate efficient business, significantly reducing mileage and its environmental footprint [42]. There are several studies that support the negative impact on $CO_2$ emission (for example: [43–47]). The growing concern about the impact of fintech on climate change, coupled with conflicting research findings, necessitates new studies examining the specific role of fintech development in shaping climate outcomes in the top five Chinese-invested African economies. We present summaries of further literature review in Table 1.

**Table 1.** Literature Review.

| No. | Authors | Countries | Data | Variables | Methodologies | Finding |
|---|---|---|---|---|---|---|
| 1 | Guo and Yin [48] | China | 1990–2022 | Fintech and $CO_2$ | NARDL | Fintech and green technology negatively affect $CO_2$ emissions in a positive shock and have positive effects in a negative shock. |
| 2 | Li et al. [49] | Belt and Road countries | 1990–2020 | Fintech and $CO_2$ | CS-ARDL | Fintech can help to reduce $CO_2$ emission in Belt and Road countries |
| 3 | Liu et al. [50] | China | 2000 to 2020 | Fintech and $CO_2$ | QARDL | Fintech has negative impact on $CO_2$ pollution |
| 4 | Sadiq et al. [51] | China | 2013 to 2022 | Fintech and $CO_2$ | different regression-based models | Development of fintech in China facilitates the reduction of carbon emissions and promotes climate quality. |
| 5 | Song and Hao [52] | China | 2000 to 2020 | Fintech and $CO_2$ | BARDL | Fintech can be one of the ways to digitalize the financial sector and ensure that state-of-the-art technology reduces the $CO_2$ emissions |
| 6 | Jian and Zhengjie [53] | China | Q1 2005 to Q4 2021 | Fintech and $CO_2$ | ARDL | Fintech has positive impact on $CO_2$ pollution |
| 7 | Sapkota and Bastola [54] | 14 Latin American countries | 1980 to 2010 | FDI and $CO_2$ | panelfixed and random effects models | FDI has positively related to pollution emissions |
| 8 | Shahbaz et al. [55] | Middle East and North African countries | 1990 to 2015 | FDI and $CO_2$ | GMM | FDI could cause increased $CO_2$ emission |
| 9 | Lee and Brahmas-rene [56] | European Union | 1988 to 2009 | FDI and $CO_2$ | panel cointegration techniques and fixed-effects models | Find that FDI has positive effect on $CO_2$ emission reduction, and every 1% increase in FDI could lead to a 0.017% decreasing in $CO_2$ emissions. |
| 10 | Khan and Hassan [8] | 141 developing economies | 2000–2021 | GDP and $CO_2$ | Method of Moment Quantiles Regression | GDP has positive impact on $CO_2$ pollution. |
| 11 | Zambrano-Monserrate [57] | OECD countries | 1970–2015 | GDP and $CO_2$ | CS-ARDL | GDP has positive impact on $CO_2$ emission |
| 12 | Shabani and Shahnazi [58] | Iran | 2002 to 2013 | GDP and $CO_2$ | Dynamic ordinary least squares (DOLS) | discovered a unidirectional causal relationship from GDP to $CO_2$. Moreover, in the short run, they observed dual causality between GDP and carbon emissions. |
| 13 | Hdom and Fuinhas [59] | Brazil | 1975–2016 | GDP and $CO_2$ | fully modified ordinary least square (FMOLS) and dynamic ordinary least squares (DOLS) | GDP has a positive impact on $CO_2$ pollution |
| 14 | Munir et al. [60] | five main Association of Southeast Asian Nations | 1980–2016 | GDP and $CO_2$ | panel test of Granger non-causality | economic growth causes unidirectionally carbon emissions in the Philippines, Malaysia, Singapore, and Thailand. |
| 15 | Xin et al. [61] | China | 1991–2020 | Unemployment and $CO_2$ | ARDL | Unemployment has a negative impact on $CO_2$ |
| 16 | Liu et al. [62] | 77 countries | 1991–2020 | Unemployment and $CO_2$ | STIRPAT model | Unemployment causes reduced $CO_2$ pollution |

Despite the growing interest in exploring the connections among Chinese investment, fintech, and sustainable development, there is a lack of consensus on empirical findings and outcomes. Additionally, the existing literature enhances our comprehension of the links between foreign investment, fintech, total energy consumption, and environmental pollution. However, the results vary, lack conclusive evidence, and raise concerns, primarily because the literature typically examines the relationship between two variables, such as foreign investment and economic growth, foreign investment and the environment, fintech and economic growth, or fintech and environmental pollution. Notably, there is a dearth of studies investigating the interactions among these variables, which constitute the foundations of sustainable development. Specifically, there is a lack of research on the top five African economies that have received Chinese investment. To address this lack of knowledge and make a valuable addition to the existing body of research, this study offers the following contributions: (a) To what extent does Chinese foreign investment support the top five Chinese-invested African economies in achieving their climate objectives? (b) How does the utilization of fintech influence the pursuit of sustainable development goals in the top five economies receiving Chinese investment in Africa? These two goals will aid in assessing the impact of Chinese foreign investment and fintech on sustainable development. In addition, the research examines the impact of economic performance, total energy consumption, and unemployment on the sustainable development of the five African economies with the highest Chinese investment.

## 3. Data and Methods

### 3.1. Data

The present study employs quarterly data for five select top China investment countries—Nigeria, South Africa, Egypt, Angola, Republic of the Congo—encompassing a 15-year-year period from 2006Q1 to 2021Q1. The inclusion of only these five countries was necessitated by the scarcity of recent and comprehensive data for the remaining African countries. The explanatory variables identified in the literature as influencing environmental quality (EQ) are Chinese investment, fintech, total energy consumption, economic performance, and unemployment. Except for Chinese investment and environmental quality, all of these variables were sourced from the World Bank Development Indicators Database (WDI). All variables were transformed into logarithmic form to alleviate the concerns of multicollinearity and autocorrelation. Table 2 describes the variables used in the study together with their measurement and source.

**Table 2.** Variable Description.

| Variable | Sign | Proxy | Source of Data |
|---|---|---|---|
| Environmental Quality | $CO_2$ | $CO_2$ emission per Capita | Statistical Review of World Energy (BP) |
| Chinese Investment | CFDI | Chinese investment in top five Chinese-invested African countries | China Global Investment Tracker Database |
| Financial Technology | FINTECH | Fixed broadband subscriptions per 100 persons | WDI |
| Economic performance | GDP | GDP per capita | WDI |
| energy consumption | ENERGY | Total energy consumption | WDI |
| Unemployment | UNM | Total percentage of unemployment | WDI |

Source: Authors' work.

### 3.2. Methods

Our quarterly dataset includes five countries and 15 years of data, so there are more years than countries. The variables might not be stable over time, but they follow a specific pattern (I(1)). The model is probably dynamic, meaning that it changes over time. In this scenario, opting for a panel-ARDL model, as formulated by [63,64], is deemed more suitable. According to these researchers, the panel-ARDL model offers advantages over alternative dynamic panel methods, such as fixed effects, instrumental variables, or GMM

estimators proposed by [65–67], and others. The key distinction lies in the panel-ARDL model's ability to avoid generating inconsistent estimates of the average parameter values, a concern, present in methods assuming identical coefficients across countries [68].

Certainly, the estimated model takes the shape of an Autoregressive Distributed Lag (ARDL) model with parameters p, q, q, . . ., q as denoted by Equation (1):

$$EQ_{it} = \sum_{j=1}^{p} a_{ij} EQ_{i,t-j} + \sum_{j=0}^{q} \delta_{ij}' X_{i,t-j} + \mu_i + \varepsilon_{it} \tag{1}$$

When we consider the vector of explanatory variables as 'X', parametrizing the model results in the following in Equation (2):

$$\Delta EQ_{it} = \varphi_t(EQ_{i,t-1} - \beta_i X_{it}) + \sum_{j=1}^{p-1} a_{ij}^* \Delta EQ_{i,t-j} + \sum_{j=0}^{p-1} \delta_{ij}^* \Delta X_{i,t-j} + \mu_i + \varepsilon_{it} \tag{2}$$

In our context, we focus on the key variables of interest, represented by $\beta_i$, which quantify the enduring influence of the explanatory factors on the proportion of environmental quality. Additionally, we consider the impact of the error correction mechanism. The remaining parameters, denoted as $\varphi_i$, pertain to the short-term coefficients in our analysis. It is important to note that the disturbances, indicated as $\varepsilon_{it}$, are assumed to be independently distributed over both time and units. These disturbances exhibit a mean of zero and a constant variance within each unit.

The panel ARDL model was utilized to examine the panel series data and ascertain the long- and short-term equilibrium in this study. This methodology is particularly effective in addressing two significant challenges in panel data analysis: endogeneity and serial correlation [69]. While serial correlation refers to the correlation of error terms across time, endogeneity occurs when the independent variables are associated with the error term. To address these difficulties, the ARDL model incorporates terms for error correction and lagged dependent variables, which successfully mitigates the problems related to endogeneity and serial correlation. The Pedroni cointegration test and the panel ARDL model were utilized in our study to investigate the short- and long-term correlations between the variables of interest [70]. One often-used method for panel data analysis that allows us to determine whether a stable long-term equilibrium exists is the Pedroni cointegration test. The model in this equation allows the parameters to differ among countries. Researchers such as [63,64] showed that the Mean Group (MG) estimator, which estimates parameters for each country and averages them, can consistently estimate these varying parameters. However, they also established that if the long-run coefficients are consistent across countries, a more efficient estimator called the Pooled Mean Group (PMG) can be used. The PMG estimator allows short-run parameters to vary by country while keeping long-run parameters consistent. To utilize these methods, the variables must follow a mixed pattern of I(1) and I(0) stationarity and the variables must exhibit cointegration for the model to be interpreted as an error correction mechanism. The next section will delve into the stationarity tests of the variables, the existence of cointegration, and the panel estimator.

## 4. Results and Discussions

### 4.1. Descriptive Statistics and Correlation Matrix

Table 3 presents the statistical and correlation matrices for selected variables. Before presenting the descriptive statistics and correlation results, we assessed the presence of seasonality in the variables using the method outlined by [71,72]. Our findings suggest that seasonality is not a significant concern, potentially due to the log transformation applied to all variables. The result of Table 3 indicates that the first part of Table 3 details the data series characteristics, including mean, standard deviation, maximum, minimum, probability, and observations. The second part of Table 3 displays the Spearman correlation matrices, revealing the magnitude and direction of relationships between variables. The analysis shows a negative correlation between LCFDI and $CO_2$ pollution, indicating LCFDI

decreases as $CO_2$ pollution increases [73]. Similarly, Fintech exhibits a negative correlation with $CO_2$ pollution. Conversely, other variables show positive correlations, meaning they increase alongside $CO_2$ pollution.

**Table 3.** Descriptive Statistics and Spearman Correlation Analysis Outcome.

| | $LCO_2$ | LCFDI | LFINTECH | LENERGY | LGDP | LUNM |
|---|---|---|---|---|---|---|
| Mean | −2.010594 | 6.367028 | −2.437957 | 3.076055 | 7.355442 | 2.069395 |
| Std. Dev. | 1.006098 | 1.086866 | 3.256684 | 1.499718 | 1.168331 | 0.660716 |
| Maximum | −0.347363 | 8.630522 | 2.294281 | 4.589651 | 9.075327 | 3.359333 |
| Minimum | −3.414933 | 4.700480 | −7.397834 | 0.985257 | 5.511580 | 1.127200 |
| Probability | 0.000015 | 0.000294 | 0.000001 | 0.000000 | 0.000000 | 0.000003 |
| Observations | 305 | 305 | 305 | 305 | 305 | 305 |
| **Spearman Correlation** | | | | | | |
| | 1.000000 | −0.161668 | 0.777615 | 0.798621 | 0.805973 | 0.798406 |
| | | 1.000000 | −0.303159 | −0.255171 | −0.109735 | −0.221367 |
| | | | 1.000000 | 0.941376 | 0.822728 | 0.840723 |
| | | | | 1.000000 | 0.760063 | 0.783050 |
| | | | | | 1.000000 | 0.822895 |
| | | | | | | 1.000000 |

Source: Authors' computation by using EViews 13.

### 4.2. Root Analysis of Variables

To ensure the suitability of variables for time series analysis and modeling, a thorough examination of their stationarity properties is undertaken. This involves conducting unit root tests, which are statistical procedures designed to determine whether a time series exhibits a trend or other non-stationary features. The ADF test indicated a mixed pattern of integration among the variables. As shown in Table 4, the ADF test reveals that some variables are integrated at level I(0), while others are integrated at the first difference I(1). To corroborate the findings of the Augmented Dickey–Fuller test (ADF) test and enhance the robustness of the results, the Phillips–Perron (PP) test is subsequently applied. The PP test, which accounts for potential serial correlation in the error terms, yields similar conclusions as the ADF test. Further testing using the PP test indicated that some variables are integrated at I(0), while the remaining variables are integrated at the first difference I(1).

**Table 4.** Panel Unit Root Test.

| Variables | Level | | 1st Difference | |
|---|---|---|---|---|
| | **ADF** | **PP** | **ADF** | **PP** |
| $LCO_2$ | 0.53 | 1.90 | −4.79 *** | −4.16 *** |
| LCFDI | −2.62 ** | −1.26 | −4.72 *** | −8.69 *** |
| LFINTECH | 1.31 | −0.35 | −2.87 *** | −4.92 *** |
| LENERGY | −2.53 *** | −0.55 *** | −5.56 *** | −4.61 *** |
| LGDP | −0.62 | −0.55 | −3.93 *** | −3.68 *** |
| LUNM | 0.26 | 0.44 | −4.52 *** | −3.91 *** |

Note: *p*-value 1% ***; 5% **. Source: Authors' computation by using EViews 13.

### 4.3. Analysis of Cointegration Test among Variables

Furthermore, to provide a more robust assessment of cointegration among the variables under investigation, we have employed two complementary panel cointegration tests: the Pedroni Residual Cointegration Test [74,75] and the Kao Residual Cointegration Test [76]. These tests will confirm whether cointegration exists among the variables. As presented in Table 5, upon confirmation of cointegration, we will proceed to a Panel ARDL (Autoregressive Distributed Lag) model to examine both short- and long-run cointegration results. The panel ARDL model is specifically designed to estimate the short- and long-run impacts of independent variables on the dependent variable.

**Table 5.** Pedroni Residual Cointegration Test and Kao Residual Cointegration Test.

| **Pedroni Residual Cointegration Test** | | | | |
|---|---|---|---|---|
| Alternative hypothesis: common AR coefs. (within-dimension) | | | | |
| | Statistic | Prob. | Weighted Statistic | Prob. |
| Panel v-Statistic | −1.284808 | 0.9006 | −1.241445 | 0.8928 |
| Panel rho-Statistic | −3.637712 | 0.0001 | −1.989486 | 0.0233 |
| Panel PP-Statistic | −3.626981 | 0.0001 | −2.096818 | 0.0180 |
| Panel ADF-Statistic | −1.028274 | 0.1519 | −1.482487 | 0.0691 |
| Alternative hypothesis: individual AR coefs. (between-dimension) | | | | |
| | Statistic | Prob. | | |
| Group rho-Statistic | −1.700383 | 0.0445 | | |
| Group PP-Statistic | −2.422137 | 0.0077 | | |
| Group ADF-Statistic | −2.111245 | 0.0174 | | |
| Kao Residual Cointegration Test | | | | |
| ADF | −1.366017 | 0.0860 | | |

Hypothesis: Co-integration exists.

*4.4. Panel Analysis of Short and Long-Term Impact of Chinese Investment on $CO_2$ Pollution in the Top Five Chinese-Invested African Economies*

Table 6 presents the results of a panel ARDL analysis, a robust econometric technique for investigating dynamic relationships between variables. This study uses this method to examine the short- and long-run impacts of Chinese investment on $CO_2$ pollution in the top five Chinese-invested African economies: Nigeria, South Africa, Kenya, Egypt, and Algeria.

**Table 6.** Panel ARDL Analysis.

| **Variable** | **Coefficient** | **Std. Error** | **t-Statistic** | **Prob. *** |
|---|---|---|---|---|
| Long Run | | | | |
| LCFDI | −0.567540 * | 0.334320 | −1.697608 | 0.0917 |
| LFINTECH | −0.186447 *** | 0.024202 | −7.703637 | 0.0000 |
| LENERGY | 0.924332 *** | 0.281195 | 3.287154 | 0.0013 |
| LGDP | −0.072135 | 0.094479 | −0.763497 | 0.4464 |
| LUNM | 0.429701 * | 0.239083 | 1.797290 | 0.0744 |
| Short Run | | | | |
| LCFDI | −0.126040 ** | 0.054140 | −2.327852 | 0.0213 |
| LFINTECH | −0.024177 *** | 0.004126 | −5.859453 | 0.0000 |
| LENERGY | −0.047829 ** | 0.022275 | −2.147188 | 0.0334 |
| LGDP | 0.101880 ** | 0.037524 | 2.715084 | 0.0074 |
| LUNM | 0.555314 ** | 0.292176 | 1.900616 | 0.0593 |

Notes: *p*-value < 0.01 ***; <0.05 **; <0.1 *. Source: Authors' computation by using EViews 13.

These countries represent a diverse range of African experiences with Chinese investment and rapid economic growth, making them relevant case studies for understanding the environmental implications of this investment surge. The findings reveal a surprising negative association between Chinese investment and $CO_2$ pollution. In the long run, a 1% increase in Chinese investment is associated with a 0.0567% decrease in $CO_2$ pollution. This suggests that Chinese investment, contrary to some concerns, may be contributing to a gradual reduction in $CO_2$ emissions in these African countries. The short-run impact is even more pronounced, with a 1% increase in investment leading to a 0.126040% reduction in $CO_2$ pollution. Several potential mechanisms could explain these findings. Chinese investment may be promoting the adoption of cleaner technologies and energy sources in Africa, or it may be leading to improvements in environmental regulations and enforcement. Additionally, it is possible that Chinese investment is shifting the composition of

African economies toward less polluting sectors, such as services. In the literature, Chinese investment in Africa has been extensively studied due to its significant economic impact. For instance, Wang, Yang and Yang [17] investigated the environmental consequences, particularly regarding $CO_2$ emissions, of such investments. Focusing on North Africa [77] and Sub-Saharan Africa [78], these studies provide valuable insights into the significant dynamics at play, laying the groundwork for understanding the broader environmental sustainability implications of foreign investment in Africa.

Furthermore, the impact of fintech on $CO_2$ pollution in the African panel is similar as we found as indicated by the ARDL panel results of Chinese investments. More specifically, a one percent increase in fintech development mitigates $CO_2$ pollution by 0.186447% and 0.024177% in the short and long run, respectively. However, the impact of total energy consumption and economic performance differs depending on the time frame. While total energy consumption has a positive impact on $CO_2$ pollution in the long run, it surprisingly has a negative impact in the short run. Similarly, economic performance has a positive impact in the long run but a negative impact in the short run. Finally, unemployment consistently has a positive impact on $CO_2$ pollution in both the short and long run [79]. All results are statistically significant according to ARDL panel results. Fintech's impact on reducing environmental impact and $CO_2$ emissions has been well-documented by numerous scholars in various countries [48–52]. Our stance is in agreement with this discovery as we analyse the impact of fintech on environmental pollution in highly invested African countries. Moreover, energy and unemployment have a positive impact on $CO_2$ pollution in the long run and economic development has a negative impact on $CO_2$ pollution in the long run. In the short-run finding, GDP and unemployment have a positive impact but energy consumption has a negative impact on $CO_2$ pollution. The finding from energy total consumption, economic growth, and unemployment supports the findings from [58–62].

The robustness DOLS analysis, presented in Table 7, revealed a nuanced relationship between various factors and $CO_2$ pollution. Interestingly, a 1% increase in Chinese investment for the top five invested countries led to a modest 0.016% decrease in emissions, suggesting the potential environmental benefits of targeted foreign investment. Fintech activity exhibited a surprisingly stronger negative impact, with a 0.061% decrease associated with a 1% increase. However, economic performance displayed a counterintuitive negative correlation with $CO_2$, while contrasting variables such as energy and unemployment surprisingly showed positive associations. These results corroborate the findings obtained using the panel ARDL model.

**Table 7.** DOLS Test Output.

| Var | Coef. | Std. Err | t-Stat | *p*-Value |
|---|---|---|---|---|
| CFDI | −0.016 *** | 0.004753 | −3.484982 | 0.0006 |
| FINTECH | −0.061 *** | 0.009353 | −6.578512 | 0.0000 |
| ENERGY | 0.824 *** | 0.066877 | 12.32628 | 0.0000 |
| GDP | −0.209 *** | 0.041203 | −5.094056 | 0.0000 |
| UNM | 0.239 *** | 0.042656 | 5.620581 | 0.0000 |

Note: *** represent 0.01 (*p*-value) significance levels. Source: Author' estimations using EViews 13.

## 5. Conclusions

This research investigates the nexus between Chinese outward foreign direct investment (OFDI) and financial technology (FINTECH) in reducing $CO_2$ pollution within the top five Chinese-invested African economies. Understanding this relationship is crucial for advancing sustainable development goals on a global scale. The study further incorporates total energy consumption, economic performance, and unemployment as additional metrics to assess their impact on environmental pollution reduction, a key pillar of long-term sustainability. The study utilized a quarterly dataset from 2006 to 2021, taking into account data availability. To address potential cross-sectional concerns, various panel unit root

tests were employed, indicating that all variables exhibit I(1) characteristics. Cointegration relationships among the variables were examined using Kao and seven Pedroni tests. Each model affirmed a stable long-run co-integration relationship through four Pedroni tests and an additional Kao test. Short and long-run coefficient estimates were derived using the Panel ARDL method, offering advantages such as enhanced performance in small sample sizes and addressing issues of serial correlation and endogeneity by incorporating leads and lags in the system. To validate the findings, a robustness check was conducted using the DOLS method.

Specifically, a 1% increase in Chinese investment was found to have reduced $CO_2$ emissions by 0.567540%. Similarly, a 1% increase in fintech was associated with a 0.186447% decrease in $CO_2$ levels. On the other hand, a 1% increase in total energy consumption was found to have a positive impact on $CO_2$ emissions, leading to an increase of 0.924332%. Interestingly, a 1% increase in economic performance was shown to have a beneficial effect on the environment, while a 1% increase in unemployment was associated with an increase in $CO_2$ emissions. The DOLS method is employed to conduct a robustness check. This research unveils the potential of Chinese investment and financial technology to curb $CO_2$ pollution in Africa's top five Chinese-invested economies, even within the context of the COP28 framework. While increased energy consumption and unemployment pose challenges to environmental progress, a 1% rise in Chinese investment and Fintech can lead to a 0.57% and 0.19% reduction in $CO_2$ emissions, respectively. Notably, economic growth also demonstrates a positive environmental impact, while unemployment drives emissions upward.

To capitalize on these findings and align with COP28 goals, policymakers should prioritize evidence-based strategies rooted in scientific insights. These strategies could include, incentivizing sustainable Chinese investment. Scientific research consistently shows that targeted financial and tax incentives, coupled with technical assistance, can effectively attract green technology and sustainable projects. By leveraging insights from ecological economics and environmental science, policymakers can design incentives that not only spur investment but also promote responsible resource management and biodiversity conservation. (2) Green Fintech Solutions: Building on research in environmental economics, policymakers can explore the potential of tailored Fintech solutions, such as crowdfunding platforms for renewable energy projects or blockchain technology for tracking carbon offsets, to address environmental challenges. Investments in green finance platforms, coupled with transparent regulations, can leverage financial innovation to accelerate the shift towards a sustainable economy. Also, addressing potential asymmetric information issues is particularly important when analyzing the function of Chinese OFDI and Fintech in lowering $CO_2$ emissions in African nations. Policymakers should consider implementing targeted legislation to address information asymmetry in the financial technology sector in order to alleviate these concerns. In order to enhance the level of disclosure and transparency for financial technology companies operating in these economies, it may be necessary to introduce new or revised rules. Policymakers can help to equalize the information asymmetry and foster a more competitive and efficient financial technology ecosystem by promoting increased openness and information accessibility. Cooperation among regulatory organizations, interested parties, and scholars is necessary to identify and resolve new challenges related to the environmental impact of Chinese investment and Fintech in Africa. Policymakers can enhance their understanding of the impact of Fintech and Chinese investment on environmental sustainability by including these discussions. This comprehensive plan emphasizes the importance of addressing systemic issues in order to effectively achieve sustainable development goals. (3) Clean Energy Transition: The scientific consensus on climate change highlights the urgency of transitioning to clean energy sources. Collaborative efforts with African governments, informed by climate science and energy research, are crucial for implementing effective energy-efficiency policies, expanding clean energy infrastructure, and promoting sustainable energy practices. (4) The Unemployment-Environment Nexus: Ecological economics emphasizes the interconnect-

edness of social and environmental challenges. Insights from scientific perspectives on sustainable development can inform the design of holistic solutions that address unemployment while also reducing environmental harm. This can be achieved through social safety nets, skills development programs, and support for social entrepreneurship. By incorporating these scientific insights, policymakers can enhance the effectiveness and sustainability of efforts to achieve the outlined goals in this study.

The findings of this study hold significant implications for the international community, emphasizing the importance of prioritizing African ownership and community involvement in sustainable development initiatives. This approach not only ensures respect for sovereignty and local development priorities but also maximizes the potential of Chinese outward investment (OFDI) and Financial Technology (Fintech) to drive positive environmental outcomes such as reducing $CO_2$ emissions. As nations worldwide strive to achieve the objectives outlined in COP28's vision for a greener future, understanding and implementing the strategies identified in this research can serve as a blueprint for fostering sustainable development on a global scale. In future research, with a specific focus on industries such as renewable energy and transportation, scholars can identify opportunities to expedite environmental sustainability within targeted economic sectors.

**Author Contributions:** Conceptualization, A.Z., M.N.N. and B.Z.; methodology, J.A.F., M.N.N. and E.N.K.N.; software, J.A.F., E.N.K.N. and S.E.A.; validation, A.Z., B.Z. and J.A.F.; formal analysis, J.A.F., S.E.A. and M.N.N.; investigation, M.N.N.; resources, E.N.K.N. and J.A.F.; data curation, S.E.A. and E.N.K.N.; writing—original draft preparation, M.N.N.; writing—review and editing, M.N.N. and E.N.K.N.; visualization, M.N.N. and S.E.A.; supervision, B.Z. and A.Z.; project administration, B.Z. and A.Z.; funding acquisition, M.N.N. and B.Z. All authors have read and agreed to the published version of the manuscript.

**Funding:** This research received no external funding.

**Institutional Review Board Statement:** Not applicable.

**Informed Consent Statement:** Not applicable.

**Data Availability Statement:** Publicly available datasets were analyzed in this study. These data can be found here: Statistical Review of World Energy, accessed on 28 December 2023 [https://www.bp.com/en/global/corporate/energy-economics/statistical-review-of-world-energy.html], China Global Investment Tracker Database, accessed on 29 December 2023 [https://www.aei.org/china-global-investment-tracker/], World Development Indicators, accessed on 28 December 2023 [https://data.worldbank.org/].

**Acknowledgments:** Emmanuel Letsyo (Assistance).

**Conflicts of Interest:** The authors declare no conflicts of interest.

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
