# Peer review of "A COP28 Perspective: Does Chinese Investment and Fintech Help to Achieve the SDGs of African Economies?"

_sustainability, doi:10.3390/su16073084_

Round 1

Reviewer 1 Report

Comments and Suggestions for Authors

This paper focuses on exploring the interplay between Chinese outward foreign direct investment, financial technology, and CO2 pollution in the top five Chinese-invested African economies. Although this article presents some important calculations, the following points should be noted:

1.      Evidence of the study’s relevance to an international audience is hard to find.

2.      The abstract is not sufficiently informative. The relevance of the paper for the international audience is not substantiated. It is recommended to substantiate the relevance of the topic and formulate a clear aim of the study so that the reader understands what the authors are trying to achieve.

3.      The discussion is poorly developed. Scientific substantiation of the results is missing. There is no clear link between the results obtained and previous studies by other researchers. It is suggested to compare the results of the authors’ research with the results of previously conducted research. Differences and/or similarities found should be identified. The findings and their implications should be discussed in the broadest context possible.

4.      The conclusions also lack the authors’ scientific insights.

5.      No specific proposals have been made. In their conclusions, the authors only mention: “By designing and implementing policies that respect African countries’ sovereignty and development priorities, engaging communities in decision-making, and continuously monitoring progress, we can collectively harness Chinese investment and Fintech as catalysts for sustainable development in Africa, moving towards a greener future aligned with COP28’s vision.”

6. Future research directions are not highlighted.

Reviewer 2 Report

Comments and Suggestions for Authors

Congratulations on your work. It has an interesting approach and is full of scientific, social, economic and political significance.

Author Response

We thank you for taking time off your busy schedule to review our research manuscript and coming out with such an objective review report. We are most grateful to you for your complement and encouragement.  You have given us the enthusiasm to want to submit more future manuscripts to Sustainability Journal for publication. Thanks and best regards.

Reviewer 3 Report

Comments and Suggestions for Authors

The current manuscript is written and presented with details in the research steps and results. Some points are required to improve or clarify.

1.         The data used in the empirical analysis are recorded quarterly. An important issue to be analysed in this case is the testing of the seasonality of the variables. Therefore reference should be made to the existence or non-existence of seasonality

2.         Descriptive statistical indicators (mean, variance) and the Pearson correlation coefficient are useful in the case of stationary time series (the lack of stationarity of a time series does not provide the possibility of estimating the mean for the mathematical expectation of the series). The Spearman correlation coefficient could be used instead of the Pearson correlation coefficient. Therefore, it is first necessary to present the results obtained from the stationarity test.

3.         Table 3 shows an indicator called "Probability". What does it represent? Is it likely to be the Probability associated with the Jarque Bera test?

4.         Although there are some research questions in the first part of the study, there is no reference to conclusions from them. It would be good to mention the answers found to the research questions.

Reviewer 4 Report

Comments and Suggestions for Authors

sustainability-2890864

COP28 Perspective: Does Chinese investment and fintech help to achieve SDGs of African Economies?

This study examines how financial inclusion, Chinese outward foreign direct investment (OFDI), and financial technology (fintech) intersect to influence effective climate action and CO2 pollution in the top five African economies invested in by China. The paper is interesting and well-structured.

1). I would suggest the authors to update their references:

a). Damrah S, Satrovic E and Shawtari FA (2022), How does financial inclusion affect environmental degradation in the six oil exporting countries? The moderating role of information and communication technology. Front. Environ. Sci. 10:1013326. doi: 10.3389/fenvs.2022.1013326

https://www.frontiersin.org/articles/10.3389/fenvs.2022.1013326/full

b). Song Gao, Yating Zhu, Muhammad Umar, Bilal Kchouri, Adnan Safi, Financial inclusion empowering sustainable technologies: Insights into the E-7 economies from COP28 perspectives, Technological Forecasting and Social Change, Volume 201, 2024 https://doi.org/10.1016/j.techfore.2023.123177

2). There are certain unaddressed issues. For instance, the increasing sophistication of financial technology services may give rise to asymmetric information problems. What policies could be implemented to mitigate information asymmetries? What regulatory measures need to be introduced or revised? I would recommend that the authors incorporate discussions on these matters in the final section of the paper. I addition, please see:

c). Daskalopoulos, E., A. Evgenidis, A. Tsagkanos and C. Siriopoulos (2016). Assessing variations in foreign direct investments under international financial reporting standards (IFRS) adoption, macrosocioeconomic developments and credit ratings. Investment Management and Financial Innovations, 13(3-2), 328-340. doi:10.21511/imfi.13(3-2).2016.05

https://www.sciencedirect.com/science/article/abs/pii/S0040162523008624

3). Please justify the choice of your modelling approach in comparison with competitive ones.

Round 2

Reviewer 1 Report

Comments and Suggestions for Authors

The changes are as requested.

 I wish your article every success.

Best regards.